

# Meta-analytic evidence that allelopathy may increase the success and impact of invasive grasses

Manya Singh and Curtis C. Daehler

Botany Graduate Program, School of Life Sciences, University of Hawai'i at Mānoa, Honolulu, Hawaii, United States

## ABSTRACT

**Background:** In the grass family, a disproportionate number of species have been designated as being invasive. Various growth traits have been proposed to explain the invasiveness of grasses; however, the possibility that allelopathy gives invasive grasses a competitive advantage has attracted relatively little attention. Recent research has isolated plant allelochemicals that are mostly specific to the grass family that can breakdown into relatively stable, toxic byproducts.

**Methods:** We conducted a meta-analysis of studies on grass allelopathy to test three prominent hypotheses from invasion biology and competition theory: (1) on native recipients, non-native grasses will have a significantly more negative effect compared to native grasses (Novel Weapons Hypothesis); (2) among native grasses, their effect on non-native recipients will be significantly more negative compared to their effect on native recipients (Biotic Resistance Hypothesis); and (3) allelopathic impacts will increase with phylogenetic distance (Phylogenetic Distance Hypothesis). From 23 studies, we gathered a dataset of 524 observed effect sizes (delta log response ratios) measuring the allelopathic impact of grasses on growth and germination of recipient species, and we used non-linear mixed-effects Bayesian modeling to test the hypotheses.

**Results:** We found support for the Novel Weapons Hypothesis: on native recipients, non-native grasses were twice as suppressive as native grasses (22% *vs* 11%, respectively). The Phylogenetic Distance Hypothesis was supported by our finding of a significant correlation between phylogenetic distance and allelopathic impact. The Biotic Resistance Hypothesis was not supported. Overall, this meta-analysis adds to the evidence that allelochemicals may commonly contribute to successful or high impact invasions in the grass family. Increased awareness of the role of allelopathy in soil legacy effects associated with grass invasions may improve restoration outcomes through implementation of allelopathy-informed restoration practices. Examples of allelopathy-informed practices, and the knowledge needed to utilize them effectively, are discussed, including the use of activated carbon to neutralize allelochemicals and modify the soil microbial community.

Corresponding author
Manya Singh, msingh@hawaii.edu

## INTRODUCTION

As a prime example of anthropogenic change, grasses have been deliberately moved by human civilizations, often to feed livestock (*D'Antonio & Vitousek, 1992*; *Fusco et al., 2021*), and their invasive spread has devastated many ecosystems (*Marshall et al., 2011*; *Wied et al., 2020*; *Kerns et al., 2020*; *Rhodes et al., 2021*; *Rayment et al., 2022*). The spread of non-native grasses can diminish native biodiversity by forming monocultures and modifying soil characteristics and nutrient cycling (*Perkins, Johnson & Nowak, 2011*; *Gibbons et al., 2017*; *Wied et al., 2020*; *Musso et al., 2021*; *Nagy et al., 2021*; *Soti & Thomas, 2021*). Non-native grasses may benefit from aspects of global change, including wildfire (*Davies et al., 2022*), drought (*Leal et al., 2021*; *Sommers, Davis & Chesson, 2022*), and nitrogen deposition (*Cione, Padgett & Allen, 2002*; *Sigüenza, Corkidi & Allen, 2006*). Non-native grass establishment can lead to increased wildfire frequency and/or intensity (*D'Antonio & Vitousek, 1992*; *Fusco et al., 2019*; *Tomat-Kelly, Dillon & Flory, 2021*; *Walker & Morgan, 2022*), and shortened fire cycles can push an ecosystem past the threshold of passive recovery (*D'Antonio, Hughes & Tunison, 2011*), which substantially increases costs of restoration and adds urgency to restoration planning in areas recently invaded by grasses.

Native (*Hierro & Callaway, 2021*), invasive (*Kalisz, Kivlin & Bialic-Murphy, 2021*) and domesticated/crop grasses (*Niculaes et al., 2018*) are reported to have allelopathic abilities. Across plant groups, allelochemicals differ in chemical structure and impart impacts through different mechanisms (*Cheng & Cheng, 2015*), but researchers have identified benzoxazinoids as allelochemicals that have been phylogenetically conserved within the Poaceae family (*Frey et al., 2009*; *Dutartre, Hilliou & Feyereisen, 2012*; *Niculaes et al., 2018*), with evidence supporting independent or convergent evolution of benzoxazinoids in some dicots (*Schullehner et al., 2008*; *Dick et al., 2012*). When considered together, the evidence of shared allelochemicals, disproportionate invasion success and impacts (*Linder et al., 2018*) and the large number of grass species, the grass allelopathy literature provides a unique opportunity to test important hypotheses in invasion biology and draw conclusions that can inform real world practices used to reduce the impacts of invasive grasses.

The aim of this meta-analysis was to test whether three key invasion biology theories are supported by studies investigating potential allelopathic abilities in grasses. First, we tested if the Novel Weapons Hypothesis (*Callaway & Aschehoug, 2000*; *Hierro & Callaway, 2003*; *Callaway & Ridenour, 2004*) was supported (on native recipient species, effect size of non-native grass < native grass). Second, we tested if the Biotic Resistance Hypothesis (*D'Antonio & Thomsen, 2004*; *Cummings, Parker & Gilbert, 2012*) was supported (for native grasses, effect size associated with non-native recipients < native recipients, assuming the native grass is an important contributor to native community resistance). Finally, we tested the hypothesis that increased phylogenetic distance is associated with increased allelopathic impact due to expected greater similarities in secondary chemicals among closer relatives and presumed resistance to self-produced allelochemicals

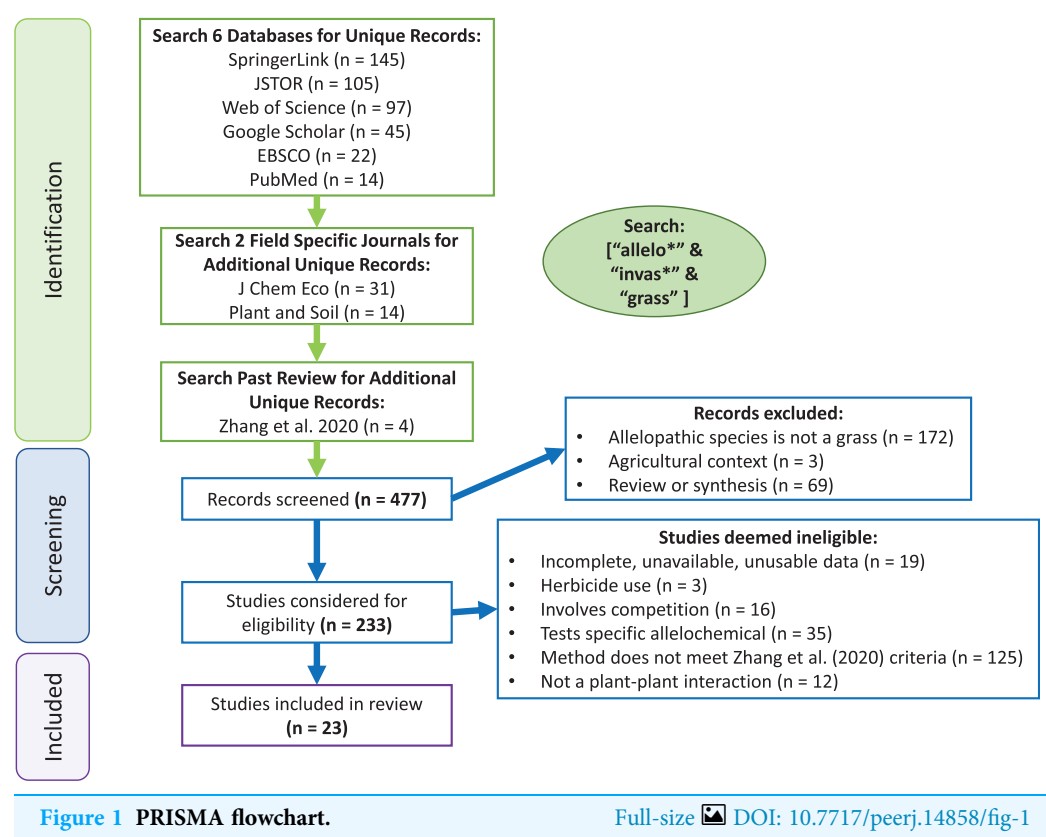

**Figure 1 PRISMA flowchart.**

(Phylogenetic Distance Hypothesis, co-efficient of smoothed phylogenetic distance <0) (*Wink, 2003*; *Zhang et al., 2020b*).

## MATERIALS AND METHODS

In our comprehensive search, three terms were used in database searches to identify studies to be included in the invasive grass allelopathy metanalysis: "invas*", "allelo*" and "grass", where "*" indicated a wildcard character. Thus, agriculture-focused research was considered only if it was presented in the context of invasion. In May 2021, multiple search engines were used to identify relevant studies for use in the meta-analysis: Web of Science, SpringerLink, EBSCO, PubMed, Google Scholar and JSTOR (Fig. 1). Specific journals were also searched: *Journal of Chemical Ecology* and *Plant and Soil* to allow searching a longer timeframe in these journals which have been historically popular for allelopathy research. Additionally, studies used in the *Zhang et al. (2020b)* meta-analysis (which included all volumes of *Allelopathy Journal*) that used grasses as the allelopathy species (species being tested for allelopathic potential) were included, but data from these studies was procured independently from each article to ensure that the methodology of extracting data remained consistent across all studies. Manya Singh performed the search strategy, and any disagreements were discussed between Manya Singh and Curt Daehler until a consensus was reached. This initial screening resulted in 477 studies, and after filtering for studies that included methodology that met criteria for inferring allelopathy (as described by *Zhang et al. (2020b)*), grass species as the source of potential allelopathic abilities

(referred to here as the 'allelopathy species'), ecological context of invasion, and separate reporting of control and test condition data with standard deviations or standard errors, 23 studies were left (*Rasmussen & Rice, 1971*; *Rice, 1972*; *Orr, Rudgers & Clay, 2005*; *Blank & Sforza, 2007*; *Barbosa, Pivello & Meirelles, 2008*; *Navarro-Cano, 2008*; *Rudgers & Orr, 2009*; *Hussain, Ahmad & Ilahi, 2010*; *Meksawat & Pornprom, 2010*; *Harnden, Macdougall & Sikes, 2011*; *Bennett, Thomsen & Strauss, 2011*; *Corbett & Morrison, 2012*; *Ghebrehiwot, Aremu & van Staden, 2014*; *Greer et al., 2014*; *Abu-Romman & Ammari, 2015*; *Ismail, Tan & Chuah, 2015*; *Perkins, Hatfield & Espeland, 2016*; *Oliveira et al., 2016*; *Jose et al., 2016*; *Uddin et al., 2017*; *Chen et al., 2018*; *Możdżeń et al., 2020*; *Guido et al., 2020*).

From each study, we collected the following information: author, year published, table/ figure where data are located, name of the allelopathy species (potentially allelopathic species), name of recipient species (species impacted by the allelopathy species), mean, standard error/deviation and sample size for both control and test conditions, lifespan of each species (annual or perennial), origin of each species, experimental method (as categorized by *Zhang et al., 2020b*), trait measured (germination or growth, for growth, aboveground preferred, then belowground, then total), duration in days, experimental environment (controlled or otherwise), condition of plant material allelochemicals were sourced from (fresh or dry), plant part used to source allelochemicals (aboveground, belowground or mixed source), dose, dose unit type, solvent and solvent polarity. Our use of 'recipient species' instead of 'test species,' which is used in other articles (including *Zhang et al. (2020b)*) to refer to the species exposed to potential allelopathy, is a change made to improve clarity around the species pairs, as across ecology, 'test species' is often used to refer to the species that is of main importance (*i.e.*, not the recipient species, but the species being tested for having or being involved in some key phenomena). Additional details about data collection and the *a priori* power calculator used prior to running the analyses are in the extended methods section (File S1).

To account for small sample bias, the delta log response-ratio (delta LRR) formula was used to calculate one "observed" effect size from each pair of control and treatment means (and standard error, sample size) (*Lajeunesse, 2015*). Two observed effect sizes were dropped because both the control and treatment mean failed the Geary check (*Lajeunesse, 2015*, standard formula), indicating that these points violated the assumption of normality. After dropping those points, we were left with a total of 524 observed effect sizes. Of the whole dataset, 23% of pairs lacked a reported dose (or information that could be used to calculate a dose), so the "mice" package was used to impute missing values based on delta LRR, standard error, and all remaining predictors in the model (*van Buuren & Groothuis-Oudshoorn, 2011*). From the "mice" function, 25 imputations were run, and for each observation missing a dose value, the median of the 25 imputed dose values was extracted for use in the modeling. Imputation *via* "mice" was done in place of the "missing values" feature included in the brms package, because dose had to be rounded and converted to a categorical variable to be used as a random effect, which is not supported by that feature. Plant species names were standardized using NCBI (*Schoch et al., 2020*). The article text and/or external sources were used to determine if each species was native (considered

locally indigenous) or non-native. Other predictors collected from each study are listed in Table S1.

In R (*R Development Core Team, 2022*), analyses utilized the 'brms' package for non-linear, mixed-effect, multi-variate Bayesian modeling (*Bürkner, 2017, 2018*), using the Student's t-distribution for the error components due to the presence of outliers. Predictors were chosen based on past evidence of significance (*Zhang et al., 2020b*) and the hypotheses to be tested. The "tree-linked" random variables refer to effects of species constrained by the phylogenetic covariance matrix, as a nested model ('phyr' package in R) (*Li et al., 2020*). The "phytools" package was used to generate the phylogenetic tree used in models (*Revell, 2012*), and the "aptg" package was used to generate a distance matrix for the full set of plant species (*Benjamin, 2017*), and the values from the distance matrix were included as a measure of phylogenetic distance in models. Phylogenetic distance was a log-scaled, smoothed term to allow for the model to inherently account for a non-linear relationship with effect size.

The non-linear model separated predictors into a "study" spline, with random effects associated with study design (study ID, nested sub-study, nested trait measured; method category, nested study duration; dose used), and a "species" spline, with random effects that capture species effects (grass and recipient species, and grass and recipient species linked to phylogenetic tree) and fixed effects for our hypotheses (origin status of grass, origin status of recipient species, phylogenetic distance). Past reviews and meta-analyses were referenced to determine which predictors were known to have correlations with allelopathic effect sizes, which we then included as random effects to account for variance (*Zhang et al., 2020b*).

To deal with the lack of independence among delta LRRs that came from the same study, the "study" spline consisted of random effects study ID (and nested variables sub study, and measured trait), dose (as a categorical variable) and experimental method (based on *Zhang et al. (2020b)* classification) (and nested variable study duration, as a categorical variable). The "species" spline consisted of random effects grass species (allelopathy species), recipient species and both species tree-linked. The fixed effects on the "species" spline were origin status of grass (hereafter, grass origin), origin status of recipient species (hereafter, recipient origin), and smoothed, log-scaled phylogenetic distance between the grass species being tested for allelopathy and the recipient species.

Prior to running the full model, an intercept model was run, which did not include any fixed effects. After generating both models, the "loo_compare" function was used to compare the fit of both models, based on both leave-one-out cross validation (LOO) and widely applicable information criteria (WAIC) values (*Vehtari, Gelman & Gabry, 2017*). The "hypothesis" function was used to test hypotheses at the 95% confidence level. Explained variance was calculated from the posterior sigma estimate (regression noise scale) and standard deviation estimates of each random effect in the intercept model. To check for publication bias, we ran a modified intercept model with log-scaled year published as a smoothed fixed effect in the "study" spline, and an Egger's regression model based on the meta-analytic residuals from the original intercept model.

**Table 1 Estimated difference and 95% CI for each hypothesis.**

|  | Estimate | 95 LCI | 95 UCI |
|---|---|---|---|
| Novel Weapons Hypothesis | −0.14 | −0.25 | −0.03 |
| Biotic Resistance Hypothesis | 0.09 | −0.01 | 0.19 |
| Phylogenetic Distance Hypothesis | −0.22 | −0.36 | −0.07 |

Note:
Negative differences were predicted *a priori* for each hypothesis test.

## RESULTS

Power analysis determined that there was sufficient power to find a difference in average allelopathic effect size, based on the number of studies and using *Zhang et al. (2020b)* as the baseline for the difference (86%, Fig. S1) (*Steidl, Hayes & Schauber, 1997*). In the intercept model, the study spline intercept was not significant (0.02, 95% CI [−0.21 to 0.25]), but the species spline intercept was significantly negative, with grasses suppressing the growth or germination of the recipient species by approximately 24% (−0.28, 95% CI [−0.52 to −0.04]). Around 35% of the variance was explained by study ID and nested variables sub-study and trait measured (15%, 8% and 11%, respectively). One-quarter of the variance was explained by method and nested variable duration (10% and 15% respectively). Another quarter of the variance was explained by grass species and recipient species (9% and 15% respectively). Dose explained 9% of the variance, meaning that only 7% of the variance in the dataset was unexplained at the observation (individual effect size) level. Phylogenetic signal from the tree-linked random effects for either the allelopathy species or the recipient species explained <1% of the variance. The Egger's test and associated contoured funnel plot of the meta-analytic residuals did not indicate significant publication bias at the $p = 0.05$ level (Fig. S2, y-intercept 95% CI [−0.03 to 0.04]). Allelopathic impacts were not significantly related to publication year (Fig. S3, y-intercept 95% CI [−0.32 to 0.14], slope 95% CI [−0.29 to 0.28]). The full model was better than the intercept model by LOO and WAIC criteria (Table S1).

The Novel Weapons Hypothesis was supported by the full model (Table 1). For native recipients, non-native grasses on average were almost twice as suppressive (24%) as native grasses (13%). The predicted average effect size of native grasses on native recipients, was weakly significantly different from zero (−0.14, 95% CI [−0.29 to 0.02], 90% CI [−0.26 to −0.01], Fig. 2). The predicted average effect size of non-native grasses on native recipients was significantly negative (−0.27, 95% CI [−0.44 to −0.09], Fig. 2).

The Biotic Resistance Hypothesis was rejected by the full model, with weakly significant support for the alternative hypothesis, that native grasses have more negative effects on native recipients compared to non-native recipients, instead of vice versa (0.09, 90% CI [0.02–0.16], Table 1). On average, native grasses suppressed native recipients 9% more compared to non-native recipients, opposite to expectations for the Biotic Resistance Hypothesis (positive model coefficient, Table 1). The predicted average effect size of native grasses on non-native recipients was not significantly different from zero (−0.05, 95% CI [−0.22 to 0.14]), with the model finding a 66% predicted probability that the average would be negative (Fig. 3).

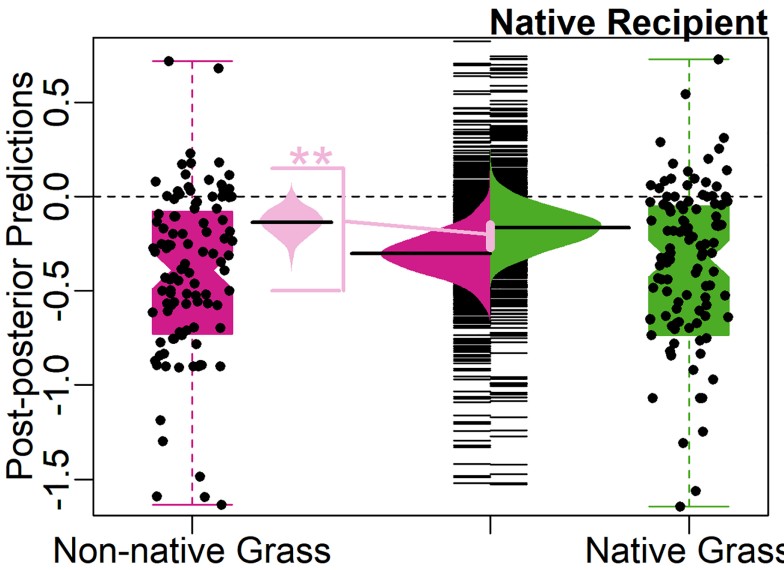

**Figure 2 Test of the Novel Weapons Hypothesis.** Center, bean plot of distribution of predicted mean effect size with long line showing the average prediction, overlayed on strip-chart of distribution of predicted population. To each side, notched boxplot, overlayed with jittered points, showing distribution of observed effect sizes. Colors represent effect of native (green, right) and non-native (magenta, left) grasses on native recipients. Center-left, bean plot of predicted difference (light pink) between average effect of native grasses and average effect of non-native grasses on native recipients, with long line showing average predicted difference. Asterisks (**) denote significance at 95% CI level.

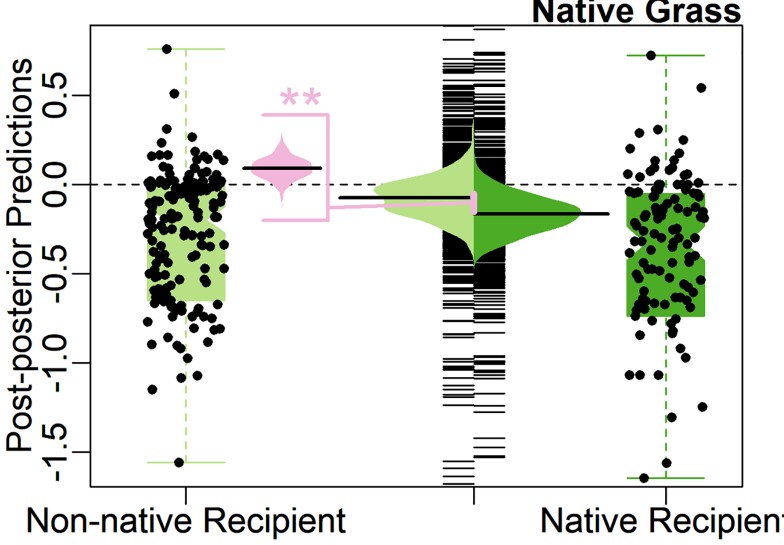

**Figure 3 Test of the Biotic Resistance Hypothesis.** Center, bean plot of distribution of predicted mean effect size with long line showing the average prediction, overlayed on strip-chart of distribution of predicted population. To each side, notched boxplot, overlayed with jittered points, showing distribution of observed effect sizes. Colors represent effect of native grasses on native (green, right) and non-native (light green, left) recipients. Center-left, bean plot of predicted difference (light pink) in average effect size of native grasses on native recipients compared to non-native recipients, with long line showing average predicted difference. Contrary to the hypotheses, natives had stronger impacts on natives than on non-natives. Asterisks (**) denote significance at 95% CI level.

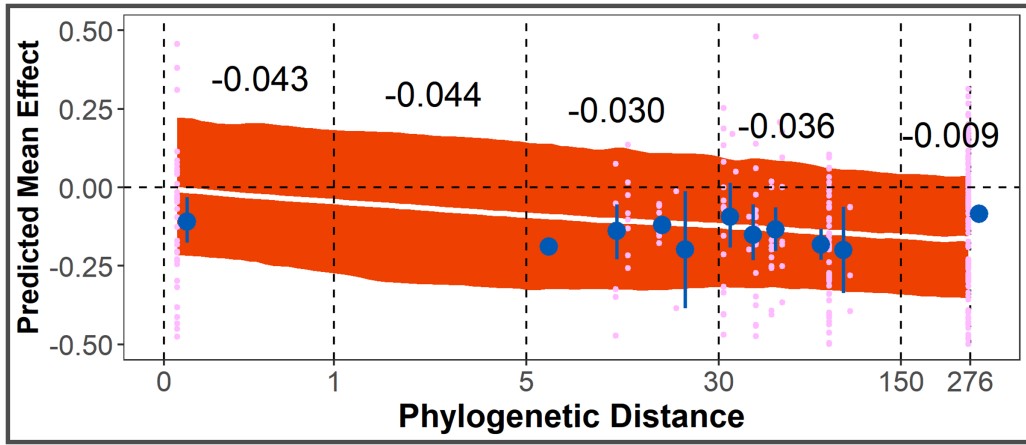

**Figure 4** **Test of Phylogenetic Distance Hypothesis.** Post-posterior predicted mean effect size (+ 95% CI) across phylogenetic distance (unitless, from distance matrix calculated using aptg package), overlayed with points representing observed effect sizes (pink) and point-ranges (in blue) representing mean + SE of observed effect sizes within y-axis bounds. Black numbers are average predicted change in effect size for that interval of phylogenetic distance.

The Phylogenetic Distance Hypothesis was supported by the full model (Table 1). There was a significant negative correlation between smoothed, log-scaled phylogenetic distance and effect size. The co-efficient of a smoothed variable cannot be interpreted directly as magnitude of change between intervals, but from model posteriors, the average allelopathic effect size for conspecific species pairs is closer to zero, compared to other species pairs with increasing phylogenetic distance (Fig. 4).

## DISCUSSION

### Support for the Novel Weapons Hypothesis and Phylogenetic Distance Hypothesis

The Novel Weapons Hypothesis (*Callaway et al., 2008*) (NWH) suggests that a lack of shared evolutionary history between non-native plants and native plants can result in allelochemical production by non-natives that has unusually large impacts on natives. We found that on a native recipient, non-native grasses are twice as suppressive as native grasses, which supports NWH. Although non-native grasses may directly release allelochemicals that have large impacts on native plants, support for NWH can also be explained by novel microbial communities associated with non-native plants, which may produce novel allelochemicals that the existing soil microbial community (recruited by native plants), has not evolved the ability to degrade (*Inderjit et al., 2011*; *Cipollini, Rigsby & Barto, 2012*). The establishment of invasive plants is generally associated with modifications to the soil bacterial community (*Torres et al., 2021*), which plays a key role in degrading allelochemicals. The identity of the microbe degrading allelochemicals may be significant if different microbes result in different by-products, and stable by-products of allelochemical degradation can be toxic (*Macías et al., 2006, 2007*; *Jilani et al., 2008*; *Hickman et al., 2021*).

In some allelopathy studies, species are studied in a reciprocal design, where each species is examined as both a potential allelopathic and recipient species. The native grasses being studied may have been chosen based on their suspected susceptibility to the soil legacy of non-native grasses, thus resulting in an over-estimation of the impact of non-native grasses. In a reciprocal design, native grasses are tested as both an allelopathic and a recipient species. Only three studies used a native grass as both an allelopathy species and the recipient species of a non-native grass (*Andropogon gerardi* in *Greer et al., 2014* and *Harnden, Macdougall & Sikes, 2011*; *Nassella pulchra* in *Chen et al., 2018*), and these points comprise just over 7% of the dataset. In a *post-hoc* analysis, we examined the predicted average allelopathic effect of native grasses *Andropogon gerardi* and *Nasella pulchra* on a native recipient species and found that the average for these grasses was more negative than the overall average (Fig. 2), suggesting that these grasses do not bias the NWH result by being less allelopathic than other grasses. Alternatively, native grasses used in studies of allelopathy may have been selected as closely related analogs of invasive species (congeneric approach, *Inderjit et al., 2008*). This type of species selection may bias allelopathic impacts downward. Less than 1% of the data consisted of a species pair where two species were of the same genus (*Eragrostis*, Fig. S4), but at the family level, 39% of the data consisted of *Poaceae* pairs. Like other analyses of the allelopathy literature (*Zhang et al., 2020b*), we found support for an increasing magnitude of allelopathic impact with increasing phylogenetic distance, but the predicted average effect size on the grass recipient species ranged from positive (ex. *Agropyron cristatum*) to negative (ex. *Eragrostis bahiensis*) (Fig. S4), indicating a high degree of variation in the overall statistical pattern of increasing allelopathic impacts with increasing phylogenetic distance. Finally, it is possible that native grasses used in allelopathic studies were chosen based on evidence of their own allelopathic abilities, against native or non-native species, seen in the field, which could result in under-estimation of the difference in impact compared to non-native grasses. Without knowing the intention of each author, it is not possible to determine how common this explanation may be, which highlights how unstated aspects of experimental design can influence our meta-analytic interpretation and understanding of important phenomena.

## Biotic Resistance Hypothesis

The Biotic Resistance Hypothesis (*D'Antonio & Thomsen, 2004*) suggests that native plants may have stronger impacts on growth and establishment of non-native plants than they do on other native plants. Although biotic resistance is generally discussed in the context of an entire native community, in native plant communities that are locally characterized by one or just a few dominant species (as is often observed in modern native grasslands), a single plant species may be the most important contributors of biotic resistance (*Prober & Lunt, 2009*; *Bennett et al., 2014*). The weapons of a native grass would be naïve to a non-native recipient species, so the lack of support for the Biotic Resistance Hypothesis suggests that a difference in mechanism or magnitude of impact of weapons may be a separating feature between grasses that have seen significant range expansion (invasive grasses), and native grasses that have been studied for allelopathy in their native range. Observations of biotic

resistance associated with some native grasses may result from other aspects of competition, such as being more resilient to stressors like drought (*Conti et al., 2018*). Additionally, it is possible that biotic resistance is reliant on soil characteristics, or the degree to which the native soil microbial community has avoided disturbance (disturbance hypothesis, *Enders et al., 2020*), which may be challenging to replicate in controlled experiments, and, potentially helping to explain the lack of evidence for the Biotic Resistance Hypothesis in our study. Finally, the greatest chance of finding evidence for the Biotic Resistance Hypothesis would be if the native species are dominants in their native communities. In general, we were not able to assess this, and therefore our study provides only a weak test of the Biotic Resistance Hypothesis.

## Variance explained by experimental design

The experimental design variables that were included as random effects in the intercept model (study/sub-study/trait, method/duration, dose, species, and tree-linked species, Figs. S4–S9) accounted for over 90% of variance in delta LRR. We included more variables as random effects compared to other meta-analyses of the allelopathy literature (*Zhang et al., 2020b*). The high level of explained variance may also be attributable to the choice of a Student's t-distribution over a Gaussian distribution for error terms, or to use of non-linear over linear formulation. One source of potential bias for the intercept model could be the imputed values for dose, as dose explained 10% of the variance in delta LRR. The magnitude of explained variance highlights the strength of Bayesian meta-analyses for mixed-effect modeling of complex, non-linear ecological phenomenon that are highly context dependent.

## Allelopathy-informed restoration practices

Based on our finding of support for NWH, in non-native grass-invaded areas, practices that account for the impact of allelochemicals may contribute to improved restoration success. Because the impact of allelopathy is dose-dependent, and the concentration of an allelochemical is influenced by soil characteristics and processes (*Kobayashi, 2004*), amendments and practices that alter these processes may result in an indirect effect on the overall allelopathic effect. For many years, activated carbon was used as a way of neutralizing or ameliorating allelochemical impacts in the field (*Callaway & Aschehoug, 2000*), but recent research suggests that in addition to a direct impact on allelochemicals, activated carbon has a broader impact on plant-soil feedback *via* modifying soil characteristics (*Lau et al., 2008*) and shifting the microbial community (*Shan et al., 2015*; *Nolan et al., 2015*). This suggests that activated carbon amendments may be useful in disrupting any dis-advantage to native plants created by soil legacy effects caused by allelopathy and altered soil feedback more generally; however it should be noted that carbon amendments do not universally benefit native plants (*Zhang et al., 2020a*), and that benefit from carbon amendments is better predicted by plant functional traits than native/invasive status (*Knauf et al., 2021*; *Cole et al., 2021*). Other options for field amendments to disrupt allelochemicals include re-conditioning the soil by growing another plant less susceptible to the allelochemicals (*Li et al., 2017*; *Schütz et al., 2019*); conducting a soil

## ALLELOPATHY INFORMED RESTORATION PRACTICES

### Microbial Inoculum

Add specific microbe, then microbe degrades allelochemicals
**Requires:**
- knowledge of **what microbe to use**
- resources and skillset to **isolate and grow microbe** to sufficient concentration for widespread use

### Activated Carbon

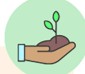

Add carbon amendment to soil, then amendment shrinks and shifts microbial community, and modifies soil characteristics, which leads to adsorbtion of allelochemicals and mitigation of allelopathic impact
**Requires:**
- resources to **purchase in bulk quantities**, price depends on source material
- context where **soil modification is beneficial**, and native species have key traits to benefit from the amendment

### Soil Transplant

Add soil and microbial community, then microbial community degrades allelochemicals
**Requires:**
- access to **source site, where significant soil collection can occur** (may only be realistic for mitigation projects)

### Outplanting

New plant recruits microbial community, then microbial community degrades allelochemicals
**Requires:**
- knowledge of **what plant to use**

**Figure 5** Four allelopathy informed restoration practices (out planting, microbial inoculum, soil transplant and activated carbon). A summary of their underlying mechanisms and what is required to utilize the practice effectively.

transplant from an area with a healthy native ecosystem or trying to reduce the concentration of allelochemicals with the addition of specific microbes *via* an inoculum approach (*Gong et al., 2018*; *He et al., 2020*; *Kheirabadi et al., 2020*). Four allelopathy-informed restoration practices are summarized in Fig. 5.

Some restoration projects in grass invaded areas have included native grasses based on their potential for resisting invasion through their functional traits (*Funk et al., 2008*) and/ or limiting similarity (*Hess et al., 2020*), but we did not find support for the Biotic Resistance Hypothesis in our analysis. Support for the Phylogenetic Distance Hypothesis does contribute to evidence supporting the limiting similarity hypothesis, assuming that more closely related species will also share traits that have been evolutionarily conserved. There are, however, concerns about the utility and practicality of basing restoration efforts on the hypothesis that limiting similarity may lead to biotic resistance, due to the challenge of determining the necessary degree of similarity, and due to the specific conditions or amount of time needed for effects of limiting similarity to act (*Hess et al., 2020*).

### Research needs for improved allelopathy-informed restoration practices

For some of the allelopathy-informed restoration practices, background knowledge is needed for the practice to be implemented successfully (Fig. 5). These "knowledge needs" point to areas where there is an urgent need for additional research. Research on the ability of specific microbes to degrade allelochemicals can contribute to the use of microbial inoculum in restoration practices. There are commercial soil amendments that include

specific microbes for improving plant growth, so research into these microbes may contribute to similar commercial products that can be specifically targeted towards grass-invaded areas. Research testing the ability of different plant species to "re-culture" grass-invaded soil is also needed, and researchers may want to prioritize testing common resilient native plants or domesticated crop species, as these species may be more accessible for use in the field. Finally, the continued use of activated carbon in a variety of contexts can contribute to an improved understanding of what contexts are appropriate for activated carbon amendments. The consideration and simulation of climate change on the efficacy of allelopathy-informed restoration practices is critical, as there is evidence that some climate events like drought can increase the potency of allelochemicals (*Borbély & Dávid, 2008*). In addition, innovative communication strategies are needed for research to have meaningful impact on restoration practices outside of academia. Platforms like the Restor Foundation's RESTOR (restor.eco) have been developed during the UN's Decade of Restoration (*United Nations, 2020*) with the aim of collecting relevant data, but practitioners may still need to invest substantial time and effort to determine the most appropriate, financially feasible practice for their context.

## CONCLUSIONS

The rise and fall of allelopathy as a trending research topic has left research gaps, but our findings supporting allelopathy as a potential mechanism that can help explain strong dominance and impact (including legacy effects) by invasive grasses. By highlighting evidence that invasive grasses may often produce allelochemicals, we hope to stimulate further research and promote consideration of allelochemical amelioration strategies after invasive grass removal, as a strategy for producing tangible improvements in conservation and restoration outcomes. It is clear that in the UN Decade of Restoration, the stakes for restoration success are high, and when it comes to the broad impacts of invasive grasses worldwide, allelopathy research presents an important opportunity to make major headway.

## ACKNOWLEDGEMENTS

This is publication #182 from the School of Life Sciences, University of Hawai'i at Mānoa.

### Funding

The authors received no funding for this work.

### Competing Interests

Curtis C. Daehler is an Academic Editor for PeerJ.

### Author Contributions

- Manya Singh conceived and designed the experiments, performed the experiments, analyzed the data, prepared figures and/or tables, authored or reviewed drafts of the article, and approved the final draft.

- Curtis C. Daehler conceived and designed the experiments, authored or reviewed drafts of the article, and approved the final draft.

## Data Availability

The raw data is available in the Supplemental File. The code and intermediate files are available on Figshare: Singh, Manya (2023): all_files. figshare. Dataset. https://doi.org/10.6084/m9.figshare.21720920.v1.

## Supplemental Information

Supplemental information for this article can be found online at http://dx.doi.org/10.7717/peerj.14858#supplemental-information.

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
