# Peer review of "Meta-analytic evidence that allelopathy may increase the success and impact of invasive grasses"

_PeerJ, doi:10.7717/peerj.14858_

## Round 0.1 · original submission · Major Revisions

Kindly address the comments of reviewers particularly those of Reviewer 2.

Reviewer 1 ·

Basic reporting

The paper by Singh and Daehler is a concise meta-analysis aiming to test some theories on the allelopathic effects of graminoids in facilitating their invasion into native plant communities.
The language of the paper is easy to follow and the style is adequate for scientific communication. There are only a few typos and minor errors (see in Additional comments).
The background and the justification of the study question are properly backed up by literature citations. Figures and tables are also OK.

Experimental design

Experimental design is OK, but it would be good to see a summary table of the species and relevant info (e.g. native and non-native ranges) involved in the meta-analysis in the main text. This would be more important for the readers than listing all involved papers as done in L72-78.

Validity of the findings

Findings seem convincing, although I encountered a potential contradictory point.
The novel weapons hypothesis was supported as non-native grasses suppressed native recipients more than native grasses. This means that if a recipient is naive for the grass, the effect will be stronger. Please notice that non-native recipients can easily be naive for native allelopathic grasses, so your finding predicts a stronger effect of the native grass on the non-native recipient than on the native recipient. This is also the expectation of the biotic resistance hypothesis. But your finding was opposite, so the first two main findings of yours contradict. Either I am wrong with this logic or you need to explain your contradictory findings. One explanation may be that non-native recipients are not necessarily "non-naive" for non-native grasses, as they may come from different regions, messing up your results. So maybe if you focus more on potential naiveness instead of nativeness, results could turn more realistic - or not, but if not, futher explanations will be needed.

Additional comments

Abstract, Results:
L4: Comma before 'Overall' should be changed to full stop.
L8: If you have a few spare words for the abstract, name some of the allelopathy-informed practices.
Main text:
L45: The aim of the study was to test the three theories on the role of alleloathy in biological invasion. The use of any method is not an aim. Please rephrase.
L65: allelopathic species sounds better for me
L90: Is there any other way for referencing a website than pasting that lengthy URL in the text?
L113: Missing space between the parenthetic parts
L147-148: 'effect size' of what? Please be more explicit.
L152-154: Too many brackets and confusing wording. Please rephrase.
L161: to by?
L167: different from
L188: Missing space between the parenthetic parts
L230-231: the story would be more complete if you could explain what can then explain the biotic resistance. Some say it depends on the functional trait composition of the recipient community. If having species with traits similar to those of the potential invador, the communitiy is more resistant.
See e.g. https://doi.org/10.1016/j.tree.2008.07.013 or https://doi.org/10.1111/1365-2664.13552
You may also make use of the findings of this one in the introduction or to link your findings to real-life community-level effects: https://doi.org/10.1111/avsc.12659

Reviewer 2 ·

Basic reporting

Dear editor and authors,

I have finished reviewing the manuscript ‘Meta-analytic evidence that allelopathy may increase the success and impact of invasive grasses’ by Singh & Daehler for publication in PeerJ.
The text is well written, following the guidelines for manuscript formatting and it is based on appropriate scientific literature. It also complies with good practices in science, by presenting professional clear English and including raw data and supplementary material.

Experimental design

The authors adopted sound methods for their meta-analysis, conducting a systematic review of literature and analyzing the secondary data they retrieved. Methods are described in details and all data is provided as an Excel spreadsheet. The file is tidy and organized, but missing values are not uniformly treated: it is included as NA in some columns (column AA, for example) but as empty cells in others (column Y). I also found an inconsistence in the field ‘X’ (column B), it is decoupled from column A from line 96 to line 128 and exhibiting an error message from this point on. The authors didn’t include their source code for the analysis, what hampers understanding by readers and is not in accordance with the politics of journal about reproducibility. I didn’t review the methodological choices regarding the statistical analysis, as I’m only barely familiar with Bayesian modelling techniques.

Validity of the findings

The main aim of the study is to evaluate, using a meta-analysis approach, the effects of allelopathy on the performance of invasive grasses. Given invasive species are one of the biggest threatens to biodiversity worldwide, the study addresses a relevant question by synthetically evaluating the effects of a mechanism by which invasive grasses might succeed and impact native plant communities.
However, despite presenting valuable results that supports allelopathy as an influential factor for the success of invasive grasses, I reckon that the hypotheses are not appropriately formulated and the findings of the study extrapolate its results. Specifically, I consider only the first hypothesis proposed by the authors to be appropriated, as allelopathy is certainly a mechanism closely related to the ‘‘novel weapon hypothesis’’. However, it is not considered as a central mechanism by which the ‘‘the biotic resistance hypothesis’’ (hypothesis 2) influence the success of invasive species. Moreover, the ‘‘biotic resistance hypothesis’’ operates at the community or ecosystem level, not at the species level. Therefore, it is conceptually wrong to test whether allelopathy effects on individual native species on invasive species support or not the ‘‘biotic resistance hypothesis’’. Finally, the authors’ third hypothesis holds correct when formulated as in lines 51-55, but I don’t agree with their interpretation that it is evidence for the ‘‘phylogenetic distance hypotheses’’ (lines 179-184). The ‘‘phylogenetic distance hypothesis’’ (also known as ‘’Darwin’s naturalization hypothesis’’) is based on eco-evolutionary histories, being more related to niche theory (resource competition) than with biotic interaction’s (interference competition) mechanisms. I suggest them to keep the results and discussion aligned with what is presented at lines 51-55 instead of discussing ‘‘phylogenetic distance hypothesis’’. In general, I recommend the authors to use recent synthetic studies about the hypotheses underlying invasions, like Enders et al. (2020, Global Ecology and Biogeography), to better formulate their own hypotheses on how allelopathy is pictured within the broader context of invasion ecology.

Additional comments

DETAILED COMMENTS

I suggest the authors to consider changing the terminology they adopted to refer to the plants on which allelopathy was measured. They call it as ‘recipient’, but this term is widely used in invasion ecology to design the community (and the native species present in it) where the invasive species enters. I suggest them to use ‘test species’ instead, as done by Zhang and collaborators (2020), to avoid confusion.

The whole discussion is correctly focused on the new weapon hypothesis. The phylogenetic distance hypothesis is barely mentioned – what emphasizes my argument that this hypothesis is not appropriate. Also, the findings for the biotic resistance hypothesis (lines 228-232) are totally expected, as the biotic resistance hypothesis is related to the diversity of the recipient community instead of the performance of individual native plant species on biotic interactions with invaders.

Line 192: It is not clear if the authors agree that this could be a mechanism explaining their results; the text could be better connected to their findings.
Lines 201-226: I suggest the authors to analyze the criteria used for the selection of native species selected for studies with reciprocal design (n = 10). This information will make the discussion more robust by allowing the authors to better explain their results in relation to the native species considered. It will also reduce speculative arguments depending on the reason why the native species were chosen.
Line 207: Figure S4 and S5 are not cited in the text so far. Please consider renaming figures so they appear sequentially.

Figure 2: The colors descripted in the figure caption don’t match the figure; native is green and invasive is magenta.

Figure 3: Please consider using the same colors as in figure 2 for consistency.

Reviewer 3 ·

Basic reporting

no comment.

Experimental design

no comment.

Validity of the findings

no comment.

Additional comments

It is meaningful to test the novel weapons hypothesis, biotic resistance hypothesis and other hypothesis associated with allelopathy by a meta-analysis. The manuscript is well organized and written. My main concern is the insufficient dataset due to the limited words.
It seems insufficient to search ‘invas*’, ‘allelo*’(Line 59). The other words associated with invasion should be included such as ‘invad*’, ‘exotic’, ‘alien’, ‘introduced’, ‘non-native’, ‘non-indigenous’ etc. In addition, other words about allelopathy should be considered such as “root exudate, leachate, secondary metabolite, secondary chemicals, water extracts, aqueous extracts” etc. In fact, some words (non-native, invad*, secondary chemicals, etc.) exist in the Ms.
Specific journals “Journal of Chemical Ecology” and “Plant and Soil” were also searched (Line 63-64). Why didn’t you search the specific journals “Allelopathy Journal”, “Weed Science” and “Weed Research”? These journals often published the articles related with the themes of the present study.
Therefore, the authors should add search words to improve dataset and re-analyzed the data.

I didn’t find most of these cited references in the list (Line 72-78).

---

## Round 0.2 · Minor Revisions

I agree with both reviewers as they suggested some minor revision before we reach at a decision. Please address all the corrections suggested by the reviewers in track changes. Figure 4 is wrongly interpreted in text which need to be corrected. Also I would suggest to go through the manuscript for minor typo errors.

Reviewer 1 ·

Basic reporting

This is a thorough revision of a formerly reviewed manuscript. I have now only a few minor comments left (see attached pdf)

Experimental design

OK

Validity of the findings

OK

Additional comments

-

Annotated reviews are not available for download in order to protect the identity of reviewers who chose to remain anonymous.

Reviewer 2 ·

Basic reporting

Dear authors,

Thank you for your rebuttal letter and the reviewed version of the manuscript and supplementary files. I evaluate you succeed in addressing the comments I have made, either accepting my suggestions or clarifying the text to strength your point when in disagreement. However, the caption of Figure 2 is still wrongly referring to the colors used in the boxplots. In addition to that, I believe figures in the Supplementary Material are named incorrectly. For example, Figure S4 does not correspond to what is described in the manuscript (line 227).
Finally, I would like to suggest the authors to consider including the paragraph in lines 267-276 at the end of the section “Allelopathy-informed restoration practices”.

Experimental design

No comment.

Validity of the findings

No comment.

Additional comments

No comment.

---

## Round 0.3 · accepted · Accept

It is hereby confirmed that authors have addressed all of the reviewers' comments. I have assessed the revision myself and feel that there is no need to invite reviewers again. I'm fully satisfied with the current revision and in my opinion, the manuscript is ready for publication. I congratulate Manya and Curtis Daehler on acceptance of their work and hope they will consider PeeJ again for future submissions.